# Accessible light-controlled knockdown of cell-free protein synthesis using phosphorothioate-caged antisense oligonucleotides

Denis Hartmann [1] & Michael J. Booth [1,2 ✉]

Controlling cell-free expression of a gene to protein with non-invasive stimuli is vital to the future application of DNA nanodevices and synthetic cells. However, little emphasis has been placed on developing light-controlled 'off' switches for cell-free expression. Light-activated antisense oligonucleotides have been developed to induce gene knockdown in living cells; however, they are complicated to synthesise and have not been tested in cell-free systems. Developing simple, accessible methods to produce light-activated antisense oligonucleotides will be crucial for allowing their application in cell-free biology and biotechnology. Here, we report a mild, one-step method for selectively attaching commercially-available photoremovable protecting groups, photocages, onto phosphorothioate linkages of antisense oligonucleotides. Using this photo-caging method, upon illumination, the original phosphorothioate antisense oligonucleotide is reformed. Photocaged antisense oligonucleotides, containing mixed phosphorothioate and phosphate backbones, showed a drastic reduction in duplex formation and RNase H activity, which was recovered upon illumination. We then demonstrated that these photocaged anti-sense oligonucleotides can be used to knock down cell-free protein synthesis using light. This simple and accessible technology will have future applications in light-controlled biological logic gates and regulating the activity of synthetic cells.

[1] Department of Chemistry, University of Oxford, Mansfield Road, OX1 3TA Oxford, UK. [2] Department of Chemistry, University College London, 20 Gordon Street, WC1H 0AJ London, UK. ✉email: m.j.booth@ucl.ac.uk

The expression of a gene to a protein is the most fundamental biological process and has been reconstituted into 'cell-free' DNA logic circuits and synthetic cells for applications in biocomputing and studying living processes in a minimal environment[1]. Controlling these cell-free systems is vital for recapitulation of complex biological pathways and the application of synthetic cells. However, most methods of control are limited as they rely on the addition of signal molecules, which will change the concentration of components when added in bulk and must pass through the membrane of a synthetic cell. The use of light to control cell-free systems is very attractive as it acts orthogonally to most cellular signals, can be applied remotely, and allows for tight spatiotemporal control[2]. Multiple technologies have been developed to turn 'on' cell-free expression using light[3–5]. Only two of these are photoreversible; by attaching a photoswitch onto the DNA template[6] and using light-sensitive proteins[7]. However, in both instances the control of both the 'on' and 'off' states was poor and the RNA transcribed was still present when transcription was terminated. It is crucial to develop light-activated 'off' switches for cell-free expression that show tight regulation and remove the already transcribed RNA. One technology that might meet this need is antisense oligonucleotides (ASOs), which degrade sequence-complementary RNA via RNase H[8]. Recently, a 'transfection-style' method was developed to attempt to insert ASOs into synthetic cells to control their behaviour[9]. Light-activated ASOs have been previously generated for gene knockdown in living cells[10–13], however, these have not been tested in cell-free systems and are not widely accessible as they are expensive and time-consuming to produce, and require extensive expertise in synthetic chemistry. The development of methods to create light-activated ASOs using single-step bioconjugation from commercial starting materials would make light-controlled cell-free expression accessible to a wide range of scientists.

A mild, chemoselective method of modifying an oligonucleotide (ON) is through the reaction of small molecule halogens with commercially available phosphorothioates. Phosphorothioates are a common ON modification where an oxygen is replaced with a sulfur atom on the phosphate backbone, which increases the stability in cellular[14,15] and cell-free environments[16]. However, they also introduce an orthogonal reactive site into DNA, which has

already been exploited to attach a large number of chemical groups post solid-phase synthesis[17–22]. To our knowledge, there is only one report so far of the attachment of photoremovable protecting groups on phosphorothioates, to control a DNAzyme[23]. However, due to a rearrangement reaction in the caging process the resulting photorelease product was a phosphate, eliminating the ability to unmask a more stable phosphorothioate oligonucleotide.

Here, we report a mild, one-step method to photocage the backbone of ASOs at defined positions using phosphorothioate (PS) backbone modifications in combination with the commercially-available halogenated photocage-precursor, 2-nitroveratryl bromide (Fig. 1). Attaching the photocages across the backbone dramatically reduced the ability of the ASO to form a duplex. The combination of reduced hybridisation ability and the steric bulk on the ASOs enabled the light-controlled RNase H-mediated cutting of targeted messenger RNA (mRNA). We then applied these PS-photocaged-ASOs to control gene knockdown in cell-free protein synthesis. The resulting caged ASOs are simple to produce, well-defined and allowed for the photorelease of phosphorothioated ASOs, making light-controlled cell-free knockdown accessible. Furthermore, the release of ASOs containing phosphorothioates will enable use with the more accessible lysate-based systems that contain nucleases.

## Results and discussion

**Screening for an active *mVenus* antisense oligonucleotide.** Initially, we screened for an RNase H-active ASO against the commonly employed fluorescent protein mVenus (mV) (Supplementary Fig. 1) and identified a sequence that provided high RNase H activity in vitro. To do this, we used the common online search tool sFold (Software for Statistical Folding of Nucleic Acids and Studies of Regulatory RNAs)[24–26] to predict antisense oligonucleotides for mRNA encoding the fluorescent protein mV. Of those, 11 were chosen across the whole mRNA and tested with RNase H against the target sequence and analysed by agarose gel electrophoresis. All 11 were active, but one oligonucleotide (starting nucleotide 220) early in the sequence was found to be particularly active. This oligonucleotide was then taken forward for this study. We also designed scramble controls and tested their RNase H activity and found that all scrambles were inactive (Supplementary Fig. 2).

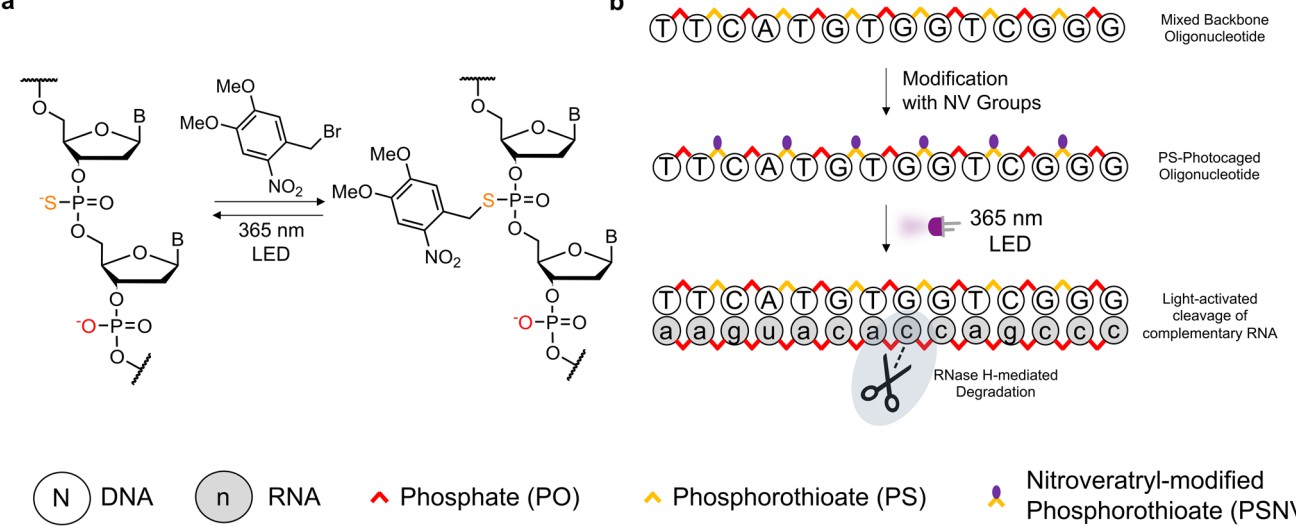

**Fig. 1 Modification of PS-containing antisense oligonucleotides and our approach to using them in controlling RNase H-mediated gene knockdown.** **a** Chemoselective modification of PS-linkages in a mixed backbone oligonucleotide with 2-nitroveratryl bromide, and subsequent uncaging following UV irradiation. **b** Caging of a mixed backbone antisense oligonucleotide with 2-nitroveratryl bromide to silence RNase H activity. UV irradiation uncaged the antisense oligonucleotide, activating RNase H-mediated target mRNA degradation.

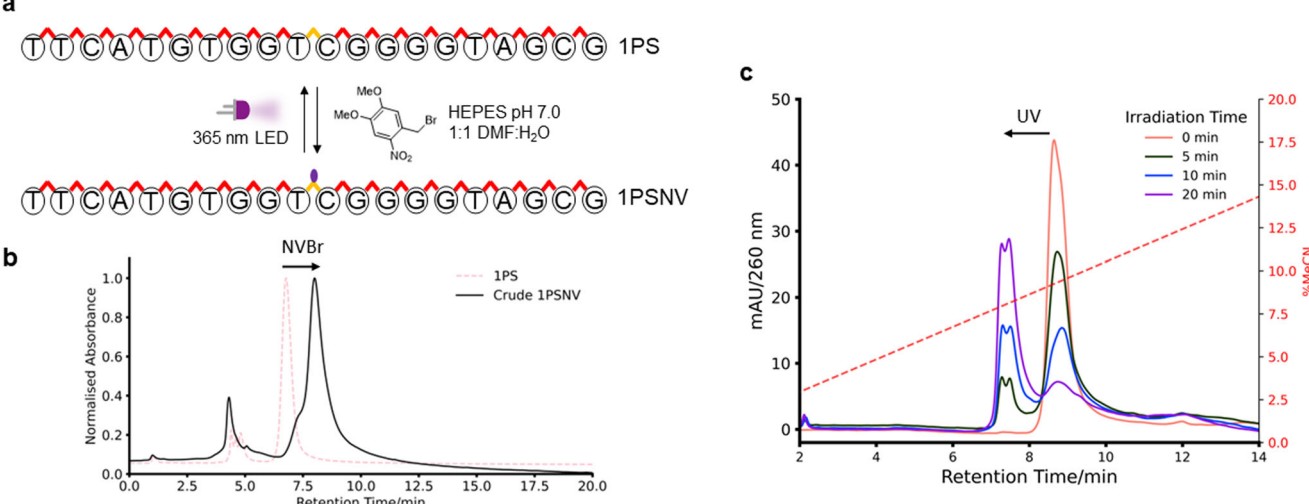

**Fig. 2 Reaction conditions for modification of PS-linkages with 2-nitroveratryl bromide and photocleavage of the reaction product. a** Optimal reaction conditions found for the modification of a PS-containing oligonucleotide and its corresponding photocleavage. **b** Crude chromatogram of the reaction after ethanol precipitation. **c** Photocleavage of the HPLC-purified 1PSNV oligonucleotide monitored by HPLC.

**Screening for selective modification conditions of PS-linkages.** The oligonucleotide starting at position 220 containing 1 PS linkage (1PS ON) at the midpoint of the sequence was then purchased, to test chemoselective caging of the PS (Fig. 2). The commonly employed small molecule photocage precursor 2-nitroveratryl bromide (NV-Br) was reacted with the 1PS ON. The ON was incubated under different buffer conditions overnight at 37 °C, precipitated with ethanol, and then analysed by high performance liquid chromatography (HPLC) (Supplementary Fig. 3). We found that with NaHCO₃, which is commonly employed to attach active esters to biomolecule amines, a large distribution of products was present due to poor chemoselectivity. Using tris buffer at pH 8, incomplete conversion was observed with a number of different products. We also analysed these reactions by liquid chromatography mass spectrometry (LCMS) (Supplementary Table 10). The desired 1x NV-modified product was observed; however, masses corresponding to more than 1 NV-group being added to the ON, as well as masses corresponding to desulfurisation to the PO oligonucleotide were also seen. This desulfurisation reaction is known for alkylated phosphorothioates, where the alkylated PS-linkage is hydrolysed to the PO linkage at pH values above 8[18].

By using HEPES buffer at neutral pH, however, high chemoselective conversion to the desired 1PSNV-modified ON was observed and could be purified by HPLC (Fig. 2a, b). To test whether the modification was at the PS-linkage, the PO-only ON was subjected to identical conditions and then analysed by LCMS after precipitation (Supplementary Table 10). The PO ON remained unchanged after the reaction, which provided proof for the chemoselectivity of the reaction to the PS-linkage.

We next tested the potential for photocleavage of the NV-modified ON and monitored it by HPLC (Fig. 2c). Efficient uncaging to the phosphorothioate starting material upon illumination with a 365 nm LED was observed. The split peak in the chromatogram is indicative of phosphorothioate diastereomers. LCMS further confirmed the production of the phosphorothioate cleavage product (Supplementary Figs. 4 and 5).

**Characterisation of different numbers of photocaged PS in oligonucleotides.** We hypothesised that to use photocaged PS-modified ONs to control gene knockdown we would require an oligonucleotide with more than 1 PS linkage. To achieve this, we purchased the 20 nucleotide (nt) long oligonucleotide with every 2nd or every 3rd PO linkage replaced with a PS linkage, resulting in oligonucleotides with 9 and 6 PS linkages in the backbone, respectively (Fig. 3a). We found that we could still chemoselectively modify the increased number of PS-linkages using the screened reaction conditions and were able to purify them by HPLC. We found an increase in retention time on HPLC, as is anticipated for the addition of a large number of hydrophobic moieties (Fig. 3b). We also monitored their UV-visible absorbance by HPLC and we saw an increase at ~360 nm with increasing amounts of NV-groups on the oligonucleotide (Fig. 3c). As expected, these 6PSNV and 9PSNV oligonucleotides showed poor solubility in purely aqueous media, and thus had to be resuspended in 25% DMSO.

To measure the effect this NV backbone modification has on oligonucleotide duplex stability, which is essential for ASO activity, melting temperatures (T_m) were recorded after incubation with the complementary strand in high salt buffer (Supplementary Table 9). Duplex stability is influenced by a number of factors, including the effective shielding of the repulsive charges between the backbones by surrounding ions as well as effective π-stacking of the bases[27]. Uncharged nucleic acids are known to still hybridise with their target reverse complementary strand[28]. It is also known that PS-linkages themselves slightly lower the $T_m$ of an oligonucleotide[14]. We measured the $T_m$ of all native and NV-modified ONs in a final concentration of 1.67% DMSO and 0.92 M NaCl, 0.92 mM, Potassium Phosphate pH 7.4 (Fig. 3d). With increasing amounts of PS-linkages, there was a small decrease in $T_m$, as expected. Upon modification with NV-groups, a drastic decrease in $T_m$ compared to the native ON was observed, with increasing reductions in $T_m$ with increasing numbers of NV groups. It was found that there is a $\Delta T_m$ of −3.71 °C for the 1PSNV, a $\Delta T_m$ of −23.03 °C for the 6PSNV and $\Delta T_m$ of −33.13 °C for the 9PSNV ONs compared to their unmodified controls. By dividing the $\Delta T_m$ of each ON by the number of NV-modifications we obtained an average $\Delta T_m$ of 3.74 °C/NV group, matching closely to the decrease observed with the 1PSNV. The presence of the NV-modification also reduced hysteresis between the heating and cooling cycles (Supplementary Figs. 25–31), which indicated a reduction in non-equilibrium secondary structures, likely due to the destabilisation of such by the modification.

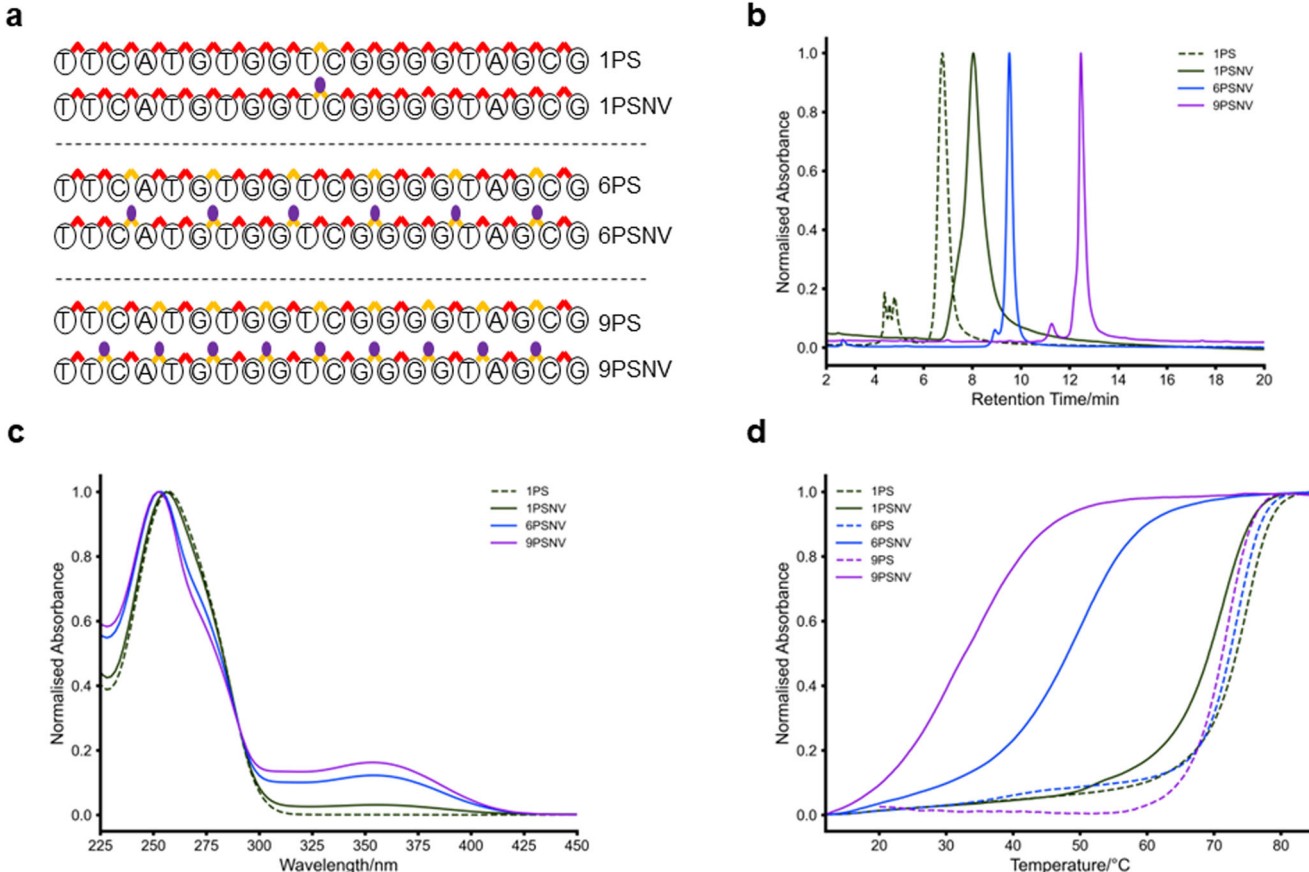

**Fig. 3 Synthesis and characterisation of NV-modified oligonucleotides containing different numbers of PS linkages. a** Modified PSNV oligonucleotides showing the pattern of 1 or 2 PO linkages between PSNV linkages, along with a singly PS-modified oligonucleotide. **b** HPLC-chromatogram of all the NV-modified oligonucleotides, compared to the 1PS oligonucleotide. A drastic increase in hydrophobicity was observed with increasing degree of NV-modification. **c** UV-visible absorbance spectra of the NV-modified oligonucleotides as recorded by HPLC. An increase in the absorbance at ~360 nm, corresponding to the NV-group is observed, as well as a slight hypsochromic shift of $\lambda_{max}$ at ~260 nm. **d** Representative melting temperature ($T_m$) curves recorded for the PS- and PSNV-oligonucleotides. A small decrease in $T_m$ was observed with introduction of more PS-linkages, and a drastic decrease in $T_m$ was observed upon modification of PS-linkages with 2-nitroveratryl-bromide.

We also synthesised the ethylated PS-linkage[22] using our screened method to compare the effect on $T_m$ (Supplementary Fig. 6). The NV-modification shows a stronger impact on $T_m$ than the ethyl (Et) group ($\Delta T_m$ of 3.71 °C for NV compared to 2.12 °C for Et), likely due to the increased hydrophobic surface of NV compared to Et.

**Control of RNase H activity using NV-modified oligonucleotides.** We next wanted to test if the reduction in duplex stability from the NV-modification on the ASOs could be used to control RNase H activity using light. RNase H has a shallow, positively charged groove, which fits the charged PO/PS backbone of the DNA in a RNA/DNA hybrid duplex[29]. Therefore, we hypothesised that if any of the NV-modified ASOs did bind the mRNA, by alkylating the PS-linkage, we would prevent RNase H from recognising its substrate as we removed the charge and increased the steric bulk of the site.

In addition to the 20nt ONs tested in Fig. 3, we screened for a truncated sequence that retained RNase H activity. This allowed for a lower number of PS-linkages over the oligonucleotide at the same density (every second PO linkage replaced), while not bearing as many NV groups for caging. We found that truncating the ASO sequence at its 3' end down to 14 nt allowed for slightly improved RNase H activity (Supplementary Fig. 7) and gene knockdown in cell-free conditions (Supplementary Fig. 8). We chose this sequence to have alternating PO and PS linkages,

which gives the same pattern as the previous 20nt-9PS ON, but with a total of only 6 PS linkages on a 14 nt ON, giving us a 14nt-6PS ON.

As we did not know how strongly the modification affects the 260 nm absorbance of the oligonucleotide, and thus accuracy of oligonucleotide concentration by common spectrophotometric techniques, we purchased the sequences with a Texas Red (TxRd) fluorophore attached at the 5' end. We increased the amount of DMF in the modification reaction to 75% to ensure solubility of the now even more hydrophobic oligonucleotide. We tracked the reaction progress of both the unlabelled and TxRd-labelled 20nt-6PS ON by agarose gel electrophoresis (Supplementary Fig. 9). The reaction was found to be complete after 2 h for both oligonucleotides with no further change observed up to 24 h. The TxRd-labelled 20nt-6PS and -9PS ONs were subjected to an increased reaction time of 3 days. This increased incubation had no impact on yield, as observed by both gel electrophoresis (Supplementary Fig. 10) as well as HPLC (Supplementary Fig. 11). Our reaction conditions allowed for largely complete modification as seen by polyacrylamide gel electrophoresis (Supplementary Fig. 12) for all three TxRd-modified ONs, and the oligonucleotides were purified by HPLC as previous, verified by UV-Vis (Supplementary Fig. 13) and purity confirmed by LCMS.

We then tested the ability of the modified ONs to control RNase H mediated degradation of mRNA. We incubated the

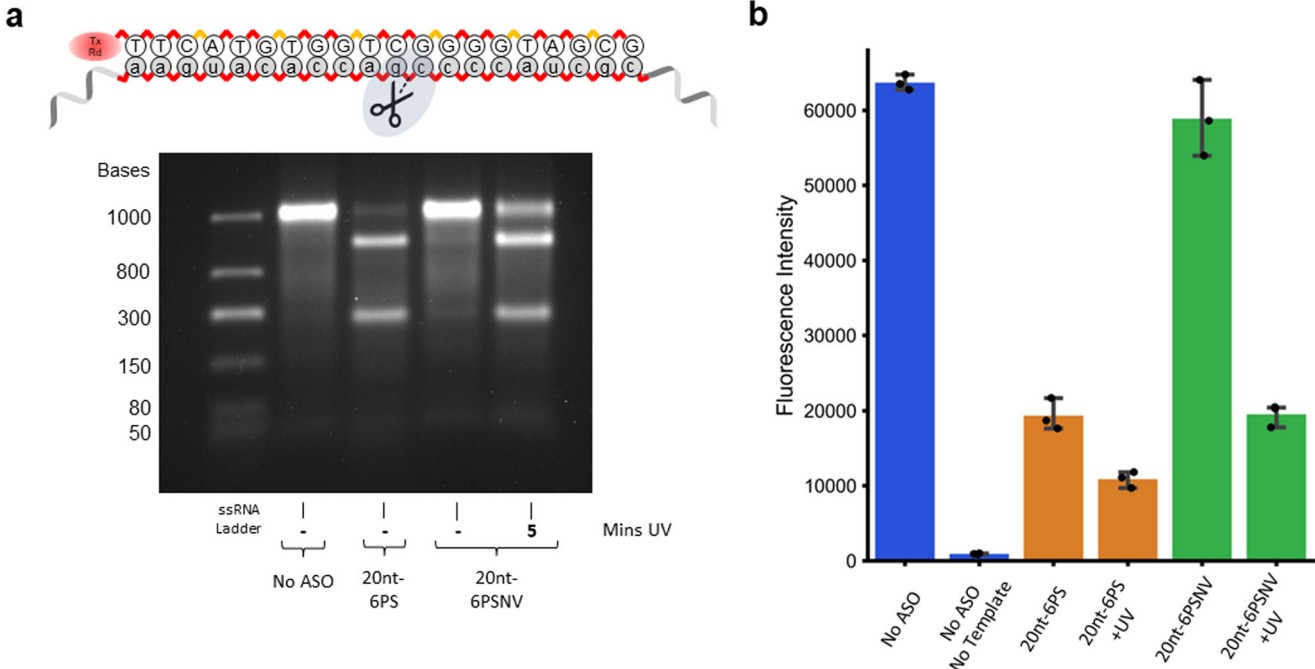

**Fig. 4 Control of RNase H activity and cell-free protein synthesis using the 20nt-6PSNV oligonucleotide. a** Agarose gel showing control of RNase H activity against the target *mVenus* mRNA by caging the 20nt-6PS ON and uncaging with UV light (0.9 eq. of ASO to mRNA). The uncropped gel can be found at Supplementary Fig. 38. **b** Cell-free protein synthesis of mVenus under different ASO and illumination conditions. When using the 20-nt 6PSNV ON, knockdown of protein synthesis was only observed following illumination. Error bars show 95% confidence interval ($n = 3$).

unmodified and NV-modified ONs with mV mRNA in the presence of RNase H for 1 h with and without illumination and analysed the mRNA degradation by agarose gel electrophoresis. For the 20nt-6PSNV ON, we were very pleased to see almost complete abolition of RNase H activity (Fig. 4a). Upon illumination with UV light for 5 min, we also see good recovery of activity. For the 20nt-9PSNV ON we also saw complete absence of RNase H activity with the modified ON (Supplementary Fig. 14). We could reactivate activity of the ASO using UV-irradiation as prior and we were also able to generate a graded response by dosing the amount of UV-light applied. Lastly, we tested RNase H activity of the truncated 14nt-6PSNV ON (Supplementary Fig. 15). Prior to irradiation, we again saw no cleavage of the mRNA by RNase H similar to the 20nt-9PSNV ON, whereas upon illumination with UV light for 5 min we observed recovery of RNase H activity.

**Application of photocaged PS antisense oligonucleotides in controlling cell-free protein synthesis.** Following demonstration of the ability to control RNase H activity with light, we wanted to apply our PSNV ONs to control cell-free protein synthesis. Light-activated ASOs have not previously been applied to cell-free protein synthesis and would add an important 'off' switch to controlling protein output. We tested our modified ONs in a commercial cell-free protein synthesis kit (PURExpress) supplemented with a DNA template encoding for mV and RNase H. The intact protein output was measured by fluorescence of mV.

Using the unmodified 20nt-6PS ON, we saw a 70% decrease in protein synthesis (Fig. 4b). Upon illumination of the system with UV light for 5 min, a further decrease in protein synthesis to 84% was observed, attributable to UV damage to the system. When using the photocaged 20nt-6PSNV, there was only a minimal decrease in protein synthesis with the caged ON of 7.6%. Upon illumination with UV light we reactivated the activity of the ASO and saw a 70% decrease in protein synthesis.

We also demonstrated that the 20nt-9PSNV-modified ON (Supplementary Fig. 16) and the 14nt-6PSNV ON (Supplementary Fig. 17) were able to control cell-free protein synthesis with light. With the 20nt-9PSNV ON, we saw a 17% residual knockdown activity in the absence of light. Following 10 min of UV irradiation (increased due to the extra number of photocages), we saw a reduction in protein yield of 66%. Using the 14nt-6PSNV ON without illumination, a 30% decrease in overall protein output was observed. This reduction was only 13% when RNase H was removed, with the residual activity likely due to steric blocking of translation. Upon illumination of the 14nt-6PSNV ON for different durations, in presence of RNase H, we achieved a graded gene knockdown response. Using 10 min of UV, the gene knockdown from the light-activated ASOs were 83% (20nt-9PSNV) and 97% (14nt-6PSNV) of the activity of the corresponding illuminated, non-modified ASOs.

For all 3 ONs tested, we also measured the TxRd-fluorescence after incubation (Supplementary Figs. 18–20). We noticed that in all cases, the fluorescence increased after photocleavage, meaning that the NV-modified ONs see some degree of fluorescence quenching, likely due to the presence of NV-groups.

To enable use of these ONs without the need for a fluorescent label, we prepared a surrogate molecule **S1** from *O,O*-diethylpho-sphorothioate, modified with 2-nitroveratryl bromide (see Chemical Synthesis Section in Supplementary Information). The molar extinction coefficient was then measured in 25% DMSO, which was used to solubilise the ONs prepared previously (Supplementary Fig. 21, 5.3 mM$^{-1}$ cm$^{-1}$ at $\lambda_{max}$ 352 nm). This will allow for future quantification of these modified ONs by measurement of the NV-absorbance on the ON.

We also tested the stability of NV-modified ONs at pH levels encountered in cell-free protein synthesis systems (PURExpress operates at 50 mM HEPES at pH 7.6)[30]. We incubated the TxRd-labelled 20nt-6PSNV ON in 50 mM HEPES pH 7.6 as well as 50 mM HEPES pH 8.0 at 37 °C for 4 h and analysed the ON by PAGE (Supplementary Fig. 22). No visible degradation was

observed at either pH over the 4 h, indicating stability of the modification at commonly encountered conditions for cell-free expression over a timescale at which synthesis usually reaches a plateau[31].

Lastly, we tested if these backbone NV modifications could improve nuclease stability, beyond what PS-linkages can provide[15] by preventing enzymes from accessing the PO-linkages between modification sites (Supplementary Fig. 23). We chose DNase I for this, as it is a highly active and ubiquitous enzyme in nature and accounts for the majority of nucleolytic activity of DNA in serum[32]. As expected, the PO-only ON showed rapid degradation, whereas the 20nt-9PS ON showed drastically improved stability. We then tested the TxRd-tagged 20nt-9PS and -9PSNV ONs (note that the pH of the gel buffer had to be adjusted to 7.5 to ensure stability of the NV-groups). We found that the NV-modified ON showed reduced degradation over the measured 6 h (Supplementary Fig. 24), thus reducing access of DNase I to the PO-linkages.

**Conclusions**. Here, we have described an accessible photoactive modification for the control of oligonucleotides for application in cell-free biology. Our light-activated oligonucleotides were produced in a single step from a commercially-available photocage with commercially-available phosphorothioate-modified DNA. The installation of the nitroveratryl groups was chemoselective for phosphorothioates in a mixed backbone oligonucleotide and could be removed with ultraviolet light, resulting in the formation of the original phosphorothioate oligonucleotide, without scarring. Multiple phosphorothioates were easily modified, resulting in diverse backbone chemistries. These photoactivatable oligonucleotides showed a drastic reduction in complementary strand binding and a reduction in solubility in aqueous media. These modified oligonucleotides were RNase H-inactive, but recovered RNase H-activity upon illumination with UV light. ASOs are underused in cell-free systems because they require transfection when knockdown is required[9], however, our light-activated ASOs only knockdown cell-free expression following illumination so can be included from the start of the experiment. Our technology is an accessible switch that will open up new possibilities for the remote control of synthetic cells and biological logic circuits.

Current limitations of this approach are the strong reduction in aqueous solubility upon modification of the oligonucleotide, which required its resuspension in 25% DMSO and the activation with UV irradiation. Future work to address these issues will include ON modification using water-soluble and more red-shifted analogues of the photocages, expanding and facilitating their use.

## Methods

**General**. All reactions were performed under ambient laboratory lighting conditions. Reagents employed were used as supplied by the manufacturers. Gel Data was analysed using ImageLab. Data was plotted using python's matplotlib and seaborn libraries. Error bars were computed using seaborn and show a 95% confidence interval.

**Polyacrylamide gel electrophoresis (PAGE)**. Gels were cast by hand with the BioRad Mini-PROTEAN® handcasting equipment at a thickness of 0.75 mm. Denaturing gels were run with 16% PAA/BisAA (19:1), 7 M Urea, 1x TBE Buffer (adjusted to pH 7.5 for PSNV oligonucleotides) or 1x HEPES:Acetate:EDTA (HAE) Buffer (40 mM HEPES, 20 mM sodium acetate, 1 mM tetrasodium EDTA, pH 7) and polymerised using 0.8% APS and 0.05% TMEDA at 250 V in 1X TBE buffer. Samples were loaded with 95% formamide, 0.015% SDS, 5 mM EDTA, 0.025% bromophenol blue and 0.025% xylene cyanol. Gels were stained with Gel-Red® nucleic acid stain (Biotium) or TxRd-dye measured without staining, or visualised by placing the gel on a TLC plate and illuminating with 254 nm light.

**Agarose gel electrophoresis (AGE)**. RNA Gels were prepared at 2% agarose in 1x TBE buffer, prestained with Sybr® Green II and run at 100 V in 1x TBE buffer. Samples were prepared using RNA loading dye (NEB B0363S) and heated to 70 °C for

10 min then cooled on ice before loading to denature the RNA. Samples were run against a low-range ssRNA ladder (NEB, N0364S) or ssRNA ladder (NEB, N0362S).

**UV illumination**. Samples were held in a PCR tube rack (StarLabs) over aluminium foil with open lids. Irradiation was then performed top-down with a ThorLabs 365 nm LED (M365L3) equipped with a collimator (COP5-A) from a distance of 34 cm at 2/6 power setting (5.65 mW·cm$^{-2}$), controlled by a ThorLabs driver (LEDD1B) set at 1 A maximum drive current.

**Preparation of linear *mVenus* template DNA**. Linear DNA encoding for the fluorescent protein mVenus was prepared by Polymerase Chain Reaction (PCR) as reported previously[3]. PCR reactions were carried out using DreamTaq DNA polymerase MasterMix (2x, ThermoFisher), Forward and Reverse Primers (Supplementary Table 1, Entry 1 and 2) at 0.25 μM concentration and 0.04 ng/μL of the HindIII-digested *mVenus* plasmid as template in a total reaction volume of 50 μL. The PCR was carried out according to the manufacturer's protocol for 35 cycles with an annealing temperature of 47 °C for 30 s, an extension time of 72 °C for 1 min (15 s/kbp) and a final extension at 72 °C for 10 min. The resulting DNA was then purified using the GeneJet PCR purification columns (ThermoFisher) following the manufacturer's protocol and eluted in 50 μL H$_2$O.

**In vitro transcription of *mVenus* mRNA**. In vitro transcription was performed using 10 ng/μL of linear *mVenus* template DNA and 2 U/μL T7 RNA polymerase (ThermoFisher) in 40 mM Tris-HCl pH 8.0, 2 mM spermidine, 10 mM DTT, 48 mM MgCl$_2$, 10 mM each of UTP, CTP, ATP and GTP in a total reaction volume of 20 μL. Samples were incubated at 37 °C for 4 h prior to addition of DNAse I (1 μL) and further incubation for 30 min. Samples were then heated to 80 °C for 20 min and the mRNA purified using the GeneJet RNA Cleanup and Concentration Micro Kit (ThermoFisher) following the manufacturer's protocol and eluted in H$_2$O.

**Screening for RNase H-active oligonucleotides against *mVenus***. The computational folding algorithm sFold[24–26] was used to identify potential antisense target sequences through input of the target mRNA. 11 sequences of 20 nucleotides in length were chosen across the whole gene and ordered as phosphate oligonucleotides, ordered by starting nucleotide in the *mVenus* gene sequence (Supplementary Table 2). The sequences were screened against 300 ng of *mVenus* mRNA (prepared as above), 6 U of RNase H (recombinant *E. coli*, Takara) and 1 ng of ASO in a buffer system containing 30 mM HEPES pH 7, 100 mM KCl, 20 mM MgCl$_2$ and 2 mM DTT in a total of 10 μL. The samples were incubated at 37 °C for 1 h, RNA loading dye (NEB, B0363S) added, samples heated to 70 °C for 10 min and then analysed by gel electrophoresis (2% agarose, 1x Tris-Borate-EDTA, 1x GelRed, 100 V). Band intensities were analysed using ImageJ by comparing band intensities for starting *mVenus* mRNA and the degradation products. Sequence 220 (Supplementary Table 2, Entry 4) was chosen for further studies.

**Ethanol precipitation**. To a 1.5 mL Eppendorf tube containing DNA in solution was added 10%vol. 3 M NaOAc pH 5.0. The tube was vortexed and centrifuged prior to addition of 2.5 volume equivalents of ice-cold ethanol. The tube was vortexed and incubated at −80 °C for a minimum of 1 h. The tube was then centrifuged at 16k RCF for 30 min at 4 °C, the supernatant removed and the resulting pellet washed with cold 70% EtOH, prior to centrifugation for another 15 min. The supernatant was then removed, the pellet dried and then resuspended.

**Screening for PS-modification conditions**. To a 1.5 mL centrifuge tube, 1PS-DNA (10 μM, Supplementary Table 3, Entry 1), the corresponding buffer (100 mM) and 2-nitroveratryl bromide (25 mM) were added in 1:1 H$_2$O:DMF in a total reaction volume of 50 μL. The reaction mixture was then vortexed, centrifuged and incubated at 37 °C overnight. The reaction mixture was then washed 3 times with chloroform to remove the majority of organic contaminants, diluted to 100 μL H$_2$O and ethanol precipitated. The oligonucleotides were then analysed and purified by HPLC on an Agilent Polaris C18 column (150 × 4.6 mm) heated to 50 °C using a gradient of 5–37% MeCN in H$_2$O over 20 min with 10 mM NH$_4$OAc pH 7–7.5 as an ion-pairing buffer throughout, prior to analysis by LCMS.

**Photocleavage of 1PSNV**. For each sample, 450 ng of 1PSNV DNA were added to a 200 μL PCR tube in a total volume of 10 μL. The samples were then irradiated with the 365 nm LED for the required time. The samples were then transferred to an HPLC injection vial and analysed on an Agilent Polaris C18 column (150 × 4.6 mm) heated to 50 °C using a gradient of 2% MeCN for 1 min, then 2–20% MeCN over 19 min with 10 mM NH$_4$OAc pH 7–7.5 as an ion-pairing buffer throughout. The photocleavage product was collected and analysed by LCMS to confirm the intact PS-linkage.

**Texas Red-modified oligonucleotide concentration measurements**. Concentration of Texas Red (TxRd)-modified nitroveratryl-oligonucleotides was determined by fluorescence. Serial dilutions of unmodified TxRd-tagged oligonucleotides were prepared at concentrations of 10, 5 and 1 μM. The concentration in ng/μL was then

determined by nanodrop, corrected for 260 nm absorbance of TxRd. 1 μL of each solution was then added to 39 μL of H$_2$O in a Corning black 384-well plate and fluorescence measurements taken on a plate reader (Tecan Infinity M1000) with Ex/Em 596/615 nm at a gain of 170 (for the 20nt-6PSNV) or gain 180 (for 20nt-9PSNV and 14nt-6PSNV) and a calibration curve generated. Concentrations of prepared samples of nitroveratryl-modified oligonucleotides were then measured by placing 1 μL of the oligonucleotide solution into 39 μL of H$_2$O in a 384-well plate and calculating the concentration from the calibration curve.

**Modification of PS-containing oligonucleotides with 2-nitroveratryl bromide**. To a 1.5 mL centrifuge tube, PS-DNA (10 μM), HEPES pH 7.0 (100 mM) and 2-nitroveratryl bromide (25 mM) were added in 1:1 H$_2$O:DMF for untagged or 1:3 H$_2$O:DMF for TxRd-tagged ONs in a total reaction volume of 50 μL. The reaction mixture was then vortexed, centrifuged and incubated at 37 °C overnight. The reaction mixture was then washed 3 times with chloroform (200 μL each, vortexed vigorously and centrifuged in a tabletop centrifuge at 14k RCF for 1 min) to remove the majority of organic contaminants, diluted to 100 μL with DMSO and transferred to an HPLC injection vial. The oligonucleotides were then purified by HPLC on an Agilent Polaris C18 column (150 × 4.6 mm) heated to 50 °C using a gradient of 5–37% MeCN in H$_2$O over 20 min with 10 mM NH$_4$OAc pH 7–7.5 as an ion-pairing buffer throughout. The DNA was resuspended in 25% DMSO analysed by LC for purity and used in downstream applications.

**Modification of 1PS oligonucleotides with ethyl iodide**. To a 1.5 mL centrifuge tube, PS-DNA (10 μM), HEPES pH 7.0 (100 mM) and 2-nitroveratryl bromide (25 mM) were added in 1:1 H$_2$O:DMF in a total reaction volume of 50 μL. The reaction mixture was then vortexed, centrifuged and incubated at 37 °C overnight. The solution was diluted to 100 μL with H$_2$O and the oligonucleotide by HPLC on an Agilent Polaris C18 column (150 × 4.6 mm) heated to 50 °C using a gradient of 5–37% MeCN in H$_2$O over 20 min with 10 mM NH$_4$OAc pH 7–7.5 as an ion-pairing buffer throughout. The DNA was resuspended in 75% DMSO analysed by LC for purity and used in downstream applications.

**Melting temperature measurements of NV-modified oligonucleotides**. To a micro-volume Quartz Suprasil$^{TM}$ cuvette with 10 mm path length and blackened sides (Hellma 105.201) was added 100 μL of buffer (1 M NaCl, 10 mM potassium phosphate 10 mM, pH 7.4). To this was added 1 μL (1 μg) of Reverse Complement DNA (100 μM, Supplementary Table 3, Entry 7) and either 1 μL of unmodified DNA (100 μM, Supplementary Table 3) or 1.6x excess of the nitroveratryl-modified DNA (to accommodate any change in 260 nm absorbance caused by the modification) as a solution in 25% DMSO. For all samples, the volume was adjusted to 120 μL total with potassium phosphate 10 mM pH 7.4 and DMSO to a final 1.6% DMSO. The oligonucleotides were then measured on a Jasco V-770 UV-Visible/NIR Spectrophotometer equipped with a Peltier temperature controller at 260 nm from 12 to 85 °C with a ramping rate of 2 °C/min, and a waiting period of 2 min at 85 °C before reversal. Melting temperatures were calculated as an average of 3 forward/reverse runs with individual melting temperatures calculated using in-built software by calculating the median of the two linear regions. Spectra were plotted using python and smoothed using a Savitzky-Golay filter.

**Light-controlled RNase H assay**. To a 200 μL PCR tube were added 300 ng of *mVenus* mRNA, 6 U of RNase H (recombinant *E. coli*, Takara), 30 mM HEPES pH 7, 100 mM KCl, 20 mM MgCl$_2$, 2 mM DTT and 5 ng of ASO (if required (0.9 eq. vs. mRNA for the 20nt-ONs or 1.6 eq. vs mRNA for the 14nt-ON) in a total of 10 μL. The resulting solutions were kept at room temperature in the dark and illuminated as required, before placing them in a thermocycler and incubated at 37 °C for 1 h. To the reactions was then added 10 μL of RNA Loading Dye (2x, NEB) and the samples heated to 70 °C for 10 min, cooled to 8 °C and analysed by AGE (2%).

**Cell-free protein synthesis**. In a 200 μL PCR tube, 10 ng/μL of *mVenus* linear template DNA were added to PURExpress® (NEB, E6800) with 0.6 U/μL RNase H (recombinant *E. coli*, Takara) and then TxRd-tagged oligonucleotides (PS or PSNV) were added to a concentration of 500 pg/μL. The % DMSO in the reactions was 1.01% for both 20nt-6PSNV and 20nt-9PSNV and 2.42% DMSO for the 14nt-6PSNV. The resulting solutions were kept at room temperature in the dark and illuminated as required, before placing them in a thermocycler and incubated at 37 °C for 4 h. 2 μL of each solution was then placed into 39 μL of H$_2$O and mixed by pipetting. 40 μL of the resulting solutions were then transferred into a 384 well plate and placed on a plate reader (Tecan Infinity M1000) and fluorescence measurements were taken for mVenus ($\lambda_{Ex/Em}$: 515/527 nm, gain 173 for 20nt-6PSNV and -9PSNV, and gain 181 for 14nt-6PSNV), as well as for TexasRed ($\lambda_{Ex/Em}$: 596/615 nm at a gain of 200). Experiments were performed in triplicates. Experimental data can be found under DOI: 10.5281/zenodo.7733943.

**Buffer stability assay**. 1 μL of a 38 μM solution of TxRd-tagged 6PS or 6PSNV ON in 50% DMSO was placed into a solution of 50 mM HEPES at either pH 7.6 or 8.0 in a total volume of 10 μL. The solutions were incubated for 4 h at 37 °C prior to analysis by denaturing PAGE in HAE buffer.

**Nuclease assay**. In a 200 μL PCR tube, 10 ng/μl of PO, PS or PSNV DNA was incubated at 37 °C with DNase I (NEB, M0303, 0.15 U/μL), DNase I Buffer (1x, NEB M0303), and up to 5% DMSO in a total volume of 40 μL. At each timepoint, a 9.5 μL aliquot was taken and diluted into 9.5 μL of 95% formamide with 10 mM HEPES pH 7 and 1% bromophenol blue and stored at 4 °C. The samples were then analysed on denaturing PAGE (7 M urea, 16%), made with TBE (ThermoFisher, pH-adjusted to 7.5 for TxRd-tagged ONs). Non-TxRd-tagged DNA was then stained using GelRed.

Oligonucleotide sequences, the synthetic procedure of compound **S1**, and NMR and MS data for compounds and modified oligonucleotides can be found in the Supporting Information.

**Reporting summary**. Further information on research design is available in the Nature Portfolio Reporting Summary linked to this article.

## Data availability

All the data generated in this study are available within the article, the Supplementary Information, and figures. Oligonucleotide sequences used within this study are found in the Supplementary Methods. Raw data for experiments relating to Fig. 4b, Supplementary Fig. 8, and Supplementary Figs. 16–20 can be found on Zenodo under https://doi.org/10.5281/zenodo.7733943.

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

## Acknowledgements

We would like to thank S. Santhakumar and R. Hicklin for technical discussions. We would also like to G. Mazzotti for experimental assistance. D.H. is grateful to the EPSRC Centre for Doctoral Training in Synthesis for Biology and Medicine (EP/L015838/1) for a studentship, generously supported by AstraZeneca, Diamond Light Source, Defence Science and Technology Laboratory, Evotec, GlaxoSmithKline, Janssen, Novartis, Pfizer, Syngenta, Takeda, UCB and Vertex. M.J.B. is supported by a Royal Society University Research Fellowship and an EPSRC New Investigator Award (EP/V030434/2).

## Author contributions

D.H. and M.J.B. conceived the project. D.H. designed, performed, and analysed the experiments, with contributions from M.J.B. D.H. and M.J.B. wrote the paper.

## Competing interests

The authors declare no competing interests.
