## [Peer Review File · Communications Chemistry]

Reviewers' comments:

Reviewer #1 (Remarks to the Author):

The study in the manuscript using phosphorothioate-bromide chemistry to construct light-controlled antisense oligonucleotides for knockdown of cell-free protein synthesis is very comprehensive, and with sufficient novelty in both chemistry and application. The chemistry of nitroveratryl (NV) modified phosphorothioate oligonucleotides was well characterized by gel electrophoresis and mass spectrum (MS), which clearly indicated the high purity of the desired products before and after UV365 light illumination. The light-controlled knockdown of cell-free protein synthesis was demonstrated with evidence in both mRNA cleavage and protein reduction, proving the effects of activating NV-modified antisense oligonucleotides by UV light. It is expected that such a system may provide artificial cells with more advanced controllable functions.

I have no question about the quality and novelty of this study, and support its publication in Communications Chemistry. The following are some potentially helpful comments for the authors to consider.

(1) Actually our research group also works on phosphorothioate-bromide chemistry for oligonucleotide modifications, and what in our experience is that, if the phosphorothioate content of oligonucleotides is high (approaching 50% or more), after hydrophobic bromide derivatization, it usually becomes somewhat hydrophobic and possibly undergoes amphiphilic self-assembly to form nano-aggregates or precipitation in neutral aqueous solution. In this case, the concentration of heavily NV-modified oligonucleotides during synthesis or application may have to be low. For example, low or sub-micromolar concentrations instead of tens or even hundreds of micromole/L. High concentrations may lead to severe yield loss during ethanol precipitation or other purification procedures. Therefore, if possible, I suggest the authors carry out some experiment to investigate whether the yield of 6 or more NV modified oligonucleotides is as high as the 1 NV modified ones after purification, and whether there is any amphiphilic self-assembly behavior for those heavily NV-modified oligonucleotides.

(2) UV light intensity (power) is an important parameter for photo-reactions. I recommend the authors to provide either a calculated or measured UV365 power (W/cm^2) applied on the solution surface.

(3) Is there any reason for the patterns of isolated phosphorothioates on the multiple phosphorothioate-containing oligonucleotides? In our experience, several consecutive modifications (such as NV) derived from consecutive phosphorothioates usually can give more significant disruption of oligonucleotide functions, compared to those with the same number but isolated modifications. Some discussion about why choosing the phosphorothioate patterns is encouraged.

(4) NV modifications on short oligonucleotides do quench the labeled fluorophores. When characterizing NV-modified oligonucleotides by gel electrophoresis, fluorescence labeling or staining is usually compromised by the quenched fluorescence. Some colorization staining methods may be more suitable, such as "Stains-All" dye (requiring higher amount of oligonucleotides to show sufficient colorization). Direct visualization of NV-modified oligonucleotides in PAGE gels by putting the gels on top of a fluorescent TLC plate and under UV254 light illumination may also work.

Reviewer #2 (Remarks to the Author):

The manuscript "Accessible light-controlled knockdown of cell-free protein synthesis using

phosphorothioate-caged antisense oligonucleotides" by Hartmann and coworkers describes a mild one-step method of alkylation of photo-cleavable protecting group selectively on the phosphorothioate backbone and demonstrates applicability of the light-controlled "off" switches in cell-free expression system. While light-controlled antisense activity is not a new concept, previous studies have only been accessible to chemists who have access to DNA synthesizers. The selective alkylation of the phosphorothioate moiety has been well-studied, but the reaction using a photo-cleavable protecting group has not been reported.

This method might be able to be performed not only by chemists but also by researchers in other fields. This method can not only be used in antisense oligonucleotides, but it may also be applicable to siRNA, guide RNA in CRISPR-Cas9, synthetic mRNAs, and any other nucleic acid technologies that involve duplex formation. The manuscript is concise and well-written. For experiment, I suggest to add some data indicating below. With this additional information, I believe the manuscript will be suitable for publication.

1) I suggest determining the molar extinction coefficient of the nitroveratryl moiety. This information will be important for readers who plan to use this method. Without it, determining the concentration by introducing the fluorophore would be necessary, diminishing the utility of the method.

2) Could you provide the time-course of the alkylation reaction. Alkylation of nine phosphorothioates was achieved in three days, indicating that modification of single/six sites may be faster. Reaction time-course give us the insight about the reaction kinetics which would be useful to use this method in the other sequences.

3) I am interested in the pH change after the alkylation reaction. In a 3-day reaction, 25 mM of 2-nitroveratryl bromide might be decomposed. I am curious to know if the buffer is enough in preventing acidification or not.

4) Could you provide the time-course stability of purified photo-caged oligonucleotide under pH 8 and pH 7 buffer. Such conditions are quite common for cell-free systems. As you mentioned, under pH8 will have desulfurization in some-extent. I feel the decomposition will be not so fast from alkylation reaction data but to judge the utility of this method, this information will be important.

5) It would be nice to have experimental data using photo-caged ASO in conjunction with an in vitro transcription/translation system. Additionally, it would be useful to have data on whether or not a complementary oligonucleotide can stop RNase H-mediated suppression.

6) Please add the paragraph about the limitation of this method

minor comments

ref 17, missing the journal name.

Reviewer #3 (Remarks to the Author):

The authors presented caging group modified antisense oligonucleotides for control the RNA degradation and cell-free protein synthesis. They introduced multiple photocaging group on the phosphate backbone to prevent duplex formation and further RNase H interaction. However, this strategy has been previously reported and the novelty is not high. Most importantly, the authors only evaluated the caged antisense oligonucleotides using RNase H activity and cell-free assay. These are too preliminary experiments and much more further study should be done at cellular level even animal

model level. In addition, the authors mentioned in conclusion ", we have described a new accessible photoactive modification for the control of oligonucleotides for application in cell-free biology and DNA nanodevices", this reviewer did not see any study about nanodevices in this manuscript.

RESPONSE TO REVIEWERS' COMMENTS

We thank the reviewers for their comments and suggestions. All changes in response to the reviewers' comments are highlighted in yellow within the updated manuscript.

Reviewer #1 (Remarks to the Author):

The study in the manuscript using phosphorothioate-bromide chemistry to construct light-controlled antisense oligonucleotides for knockdown of cell-free protein synthesis is very comprehensive, and with sufficient novelty in both chemistry and application. The chemistry of nitroveratryl (NV) modified phosphorothioate oligonucleotides was well characterized by gel electrophoresis and mass spectrum (MS), which clearly indicated the high purity of the desired products before and after UV365 light illumination. The light-controlled knockdown of cell-free protein synthesis was demonstrated with evidence in both mRNA cleavage and protein reduction, proving the effects of activating NV-modified antisense oligonucleotides by UV light. It is expected that such a system may provide artificial cells with more advanced controllable functions.

I have no question about the quality and novelty of this study, and support its publication in Communications Chemistry. The following are some potentially helpful comments for the authors to consider.

(1) Actually our research group also works on phosphorothioate-bromide chemistry for oligonucleotide modifications, and what in our experience is that, if the phosphorothioate content of oligonucleotides is high (approaching 50% or more), after hydrophobic bromide derivatization, it usually becomes somewhat hydrophobic and possibly undergoes amphiphilic self-assembly to form nano-aggregates or precipitation in neutral aqueous solution. In this case, the concentration of heavily NV-modified oligonucleotides during synthesis or application may have to be low. For example, low or sub-micromolar concentrations instead of tens or even hundreds of micromole/L. High concentrations may lead to severe yield loss during ethanol precipitation or other purification procedures. Therefore, if possible, I suggest the authors carry out some experiment to investigate whether the yield of 6 or more NV modified oligonucleotides is as high as the 1 NV modified ones after purification, and whether there is any amphiphilic self-assembly behavior for those heavily NV-modified oligonucleotides.

We thank the reviewer for this helpful comment. We have indeed found precipitation in purely aqueous media, particularly during pre-purification with chloroform, prior to HPLC purification, where a pellet is visible. We have attached images of this for the case of the 9PS oligonucleotide with the TxRd-label after the modification reaction. To ensure solubility of the heavily modified oligonucleotide we add DMSO as mentioned in the supporting information and, which as seen below, helps it to go into solution. We then use the 50% DMSO solution pictured to inject onto HPLC, and then resuspend the purified oligonucleotide in 25% DMSO after lyophilisation to ensure full solubility.

Crude TxRd-20nt-9PSNV ON
in pure water
Blue-purple pellet visible

Addition of 1 vol. DMSO
solubilises the ON

We have also measured a sample of our TxRd-labelled 6PSNV oligonucleotide in 50% DMSO with DLS, and found there to be no aggregates, as indicated by the correlation coefficient trace below.

As we see predictable linear reductions in the T_M with increasing numbers of NV, using very low % DMSO, we believe there to be no significant aggregation in these conditions either. In cell-free protein synthesis experiments, where we dilute to low concentrations of ASO and DMSO, there also appears to be no issue, as the reviewer highlights themselves when using low modified-PS concentrations.

Concerning the yield, we are afraid this will be difficult to measure, as seen in the new gel included for both the TxRd-labelled 6PSNV and 9PSNV reaction products, there are bands corresponding to not completely modified oligonucleotide present. Also, because the purification for the 1PSNV and 6PSNV methods differ, in that the 6PSNV is only washed with chloroform and then HPLC-purified, whereas the 1PSNV was directly precipitated from solution prior to HPLC, it would be impossible to have a direct comparison in yield. From what is observed in the HPLC chromatogram for the 1PSNV, however, is essentially complete modification. Whereas gel data for the 6PS and 9PS TxRd-labelled ONs (Supplementary Figures 10 and 12) clearly shows some incompletely modified intermediates, which do not appear as distinct peaks on HPLC.

(2) UV light intensity (power) is an important parameter for photo-reactions. I recommend the authors to provide either a calculated or measured UV365 power (W/cm^2) applied on the solution surface.

We thank the reviewer for this recommendation. We have measured the power of the 365 nm LED used, using a power meter, and have found the irradiance to be $5.65 \text{ mW}\cdot\text{cm}^{-2}$. This has been added to the Supporting Information.

(3) Is there any reason for the patterns of isolated phosphorothioates on the multiple phosphorothioate-containing oligonucleotides? In our experience, several consecutive modifications (such as NV) derived from consecutive phosphorothioates usually can give more significant disruption of oligonucleotide functions, compared to those with the same number but isolated modifications. Some discussion about why choosing the phosphorothioate patterns is encouraged.

We chose this pattern as the interacting motif of RNase H is through 2 phosphates (doi: 10.1093/nar/gkab1064). So by interspersing it and having the oligonucleotide evenly modified, any individual part should be prevented from being RNase H active. We also hoped this pattern might cause less aggregation than having the modifications focussed on one part of the oligonucleotide, which would result in an amphiphilic copolymer.

(4) NV modifications on short oligonucleotides do quench the labeled fluorophores. When characterizing NV-modified oligonucleotides by gel electrophoresis, fluorescence labeling or staining is usually compromised by the quenched fluorescence. Some colorization staining methods may be more suitable, such as "Stains-All" dye (requiring higher amount of oligonucleotides to show sufficient colorization). Direct visualization of NV-modified oligonucleotides in PAGE gels by putting the gels on top of a fluorescent TLC plate and under UV254 light illumination may also work.

We thank the reviewer for this helpful suggestion, as we struggled with visualisation and will be taking this into account for further studies using such modified oligonucleotides.

We have tested the suggested approach using a TLC plate by analysing the crude 6PSNV reaction using a buffer system we recently came across (HEPES:Acetate:EDTA, DOI: 10.1016/0378-1097(92)90412-h) and the band corresponding to the 6PSNV oligonucleotide is clearly visible without any stain and this data has been included in Supplementary Figure 9.

For quantification, we have now also prepared the diethylphosphorothioate with the NV group as a surrogate for the PSNV linkage, to measure its extinction coefficient upon request of Reviewer 2. This can be used as alternative for measuring concentrations of short oligonucleotides by measuring the absorbance at the λ_{\max} of 352 nm, which does not require any dye label in the future.

Reviewer #2 (Remarks to the Author):

The manuscript "Accessible light-controlled knockdown of cell-free protein synthesis using phosphorothioate-caged antisense oligonucleotides" by Hartmann and coworkers describes a mild one-step method of alkylation of photo-cleavable protecting group selectively on the phosphorothioate backbone and demonstrates applicability of the light-controlled "off" switches in cell-free expression system. While light-controlled antisense activity is not a new concept, previous studies have only been accessible to chemists who have access to DNA synthesizers. The selective alkylation of the phosphorothioate moiety has been well-studied, but the reaction using a photo-cleavable protecting group has not been reported.

This method might be able to be performed not only by chemists but also by researchers in other fields. This method can not only be used in antisense oligonucleotides, but it may also be applicable to siRNA, guide RNA in CRISPR-Cas9, synthetic mRNAs, and any other nucleic acid technologies that involve duplex formation. The manuscript is concise and well-written. For experiment, I suggest to add some data indicating below. With this additional information, I believe the manuscript will be suitable for publication.

We thank the reviewer for highlighting the strengths of our modification approach, in being easily accessible to a wide range of scientists without the necessary expertise in synthetic chemistry, as well as its potential future applicability on other nucleic acids.

1) I suggest determining the molar extinction coefficient of the nitroveratryl moiety. This information will be important for readers who plan to use this method. Without it, determining the concentration by introducing the fluorophore would be necessary, diminishing the utility of the method.

Upon this suggestion, we have synthesised the nitroveratryl-modified diethylphosphorothioate **S1** as a surrogate for the PSNV linkage in the DNA to measure the extinction coefficient, which is $5.3 \text{ mM}^{-1} \text{ cm}^{-1}$ at 352 nm (λ_{max}) in 25% DMSO (Supplementary Figure 21) and have added this to the main text (Lines 260-265).

2) Could you provide the time-course of the alkylation reaction. Alkylation of nine phosphorothioates was achieved in three days, indicating that modification of single/six sites may be faster. Reaction time-course give us the insight about the reaction kinetics which would be useful to use this method in the other sequences.

We have performed an experiment to look at the time course of the reaction with 6PS linkages, using the gel visualisation approach suggested by Reviewer 1, for the unlabelled ON as well as the TxRd-labelled ON and included this data as Supplementary Figure 9. We found the reaction to have already formed the desired product after 2 hours. We also performed the reaction for both the TxRd-6PS and -9PS and found no difference in reaction progress after 1 day between the 6PS and 9PS ON as seen in Supplementary Figure 10. We also noticed no detrimental effect upon longer incubation for up to 3 days on the yield of the reaction as seen in the HPLC data in Supplementary Figure 11. We have amended the manuscript to reflect this (Lines 201-209).

3) I am interested in the pH change after the alkylation reaction. In a 3-day reaction, 25 mM of 2-nitroveratryl bromide might be decomposed. I am curious to know if the buffer is enough in preventing acidification or not.

Upon suggestion of the reviewer, we have measured the pH change of a sample reaction without the oligonucleotide at 100 mM and 500 mM buffer concentration. The buffer used had a pH of 6.89 at 1 M. The 100 mM buffer reaction changed from pH 6.85 to 5.65, whereas the 500 mM buffer changed from 6.96 to 6.84 over the course of the reaction. This clearly shows acidification over the time course of the reaction, which can be prevented through increasing the buffer concentration to 500 mM, should this be a concern to the oligonucleotide in question. We have added this to the Supplementary Information as Supplementary Table 8.

In our case, we have not noticed any detrimental effect of this acidification on the reaction and the resulting oligonucleotide product. As we found for the point above, the reaction shows no additional progress after 1 day, nor any discernible degeneration over the following two days, showing no downside to longer incubation.

4) Could you provide the time-course stability of purified photo-caged oligonucleotide under pH 8 and pH 7 buffer. Such conditions are quite common for cell-free systems. As you mentioned, under pH8 will have desulfurization in some-extent. I feel the decomposition will be not so fast from alkylation reaction data but to judge the utility of this method, this information will be important.

We thank the reviewer for this suggestion. The PURExpress system that was used herein operates at 50 mM HEPES pH 7.6 (DOI: 10.1038/90802). For the time course of the expression, we have not seen any detrimental effect at this pH. To explicitly test this, however, we have taken the modified TxRd-labelled 6PSNV oligonucleotide and incubated it at 50 mM HEPES at both pH 7.6 and 8.0 for 4 hours at 37 °C, after which cell-free expression is usually complete and analysed it by polyacrylamide gel electrophoresis. We did not observe any degradation at either pH, showing stability at the tested pH values for the duration of the experiment, as seen in Supplementary Figure 22.

With the potential instability in mind however, we suggest long-term storage of the modified oligonucleotides at relatively neutral pH.

5) It would be nice to have experimental data using photo-caged ASO in conjunction with an in vitro transcription/translation system. Additionally, it would be useful to have data on whether or not a complementary oligonucleotide can stop RNase H-mediated suppression.

We would like to clarify that the cell-free protein synthesis data in Figure 4 and Supplementary Figures 16 and 17 were performed using the commercial in vitro transcription/translation system PURExpress. Therefore, we have already applied these photo-caged ASOs with an in vitro transcription/translation system.

Regarding the second point, due to the drastic reduction in T_m of the modified ON, a reverse complementary sequence would not bind well prior to photouncaging. Therefore, both the photocaged ASO and reverse complement strand would have to be added individually and a large excess of the reverse complementary strand used to out compete binding to the mRNA following illumination. Therefore, while the complementary strand approach has been used with other systems, we do not believe it would be a useful addition to this technology.

6) Please add the paragraph about the limitation of this method

We have amended the manuscript to include this suggestion, discussing the limitations of this method, as well as potential future avenues to circumvent this (Lines 302-308).

minor comments

ref 17, missing the journal name.

We thank the reviewer for pointing this out. This has now been amended.

Reviewer #3 (Remarks to the Author):

The authors presented caging group modified antisense oligonucleotides for control the RNA degradation and cell-free protein synthesis. They introduced multiple photocaging group on the phosphate backbone to prevent duplex formation and further RNase H interaction. However, this strategy has been previously reported and the novelty is not high. Most importantly, the authors only evaluated the caged antisense oligonucleotides using RNase H activity and cell-free assay. These are too preliminary experiments and much more futher study should be done at cellular level even animal model level. In addition, the authors mentioned in conclusion ", we have described a new accessible photoactive modification for the control of oligonucleotides for application in cell-free biology and DNA nanodevices", this reviewer did not see any study about nanodevices in this manuscript.

We believe the Reviewer has greatly underestimated the interest in cell-free expression systems and the real need to generate simple tools to control them, as outlined by ourselves in the manuscript and by Reviewers 1 and 2. These simple light-activated antisense oligonucleotides will bring the ability to deactivate cell-free protein synthesis into the hands of non-chemists, dramatically increasing their future applicability. We understand that our method might also be applied to living systems and believe this is an exciting future possibility, along with the possibility of applying it to alternative nucleic acid technologies, as mentioned by Reviewer 2.

As per comments from the Editor, we have removed the reference to DNA nanodevices.

REVIEWERS' COMMENTS:

Reviewer #1 (Remarks to the Author):

My previous comments about the amphiphilicity, UV light power and phosphorothioate patterns are all well addressed in this revised manuscript. Therefore I support its publication in Communication Chemistry.

Reviewer #2 (Remarks to the Author):

I am pleased to see that all points raised by the three referees in the 1st round peer-review have been well addressed in both the point-to-point letter and the revised version. I would like to recommend publishing the manuscript as it stands now.